# An evolutionarily conserved mechanism for cAMP elicited axonal regeneration involves direct activation of the dual leucine zipper kinase DLK

Yan Hao[1], Erin Frey[2], Choya Yoon[3], Hetty Wong[4], Douglas Nestorovski[1], Lawrence B Holzman[4], Roman J Giger[3,5], Aaron DiAntonio[2], Catherine Collins[1]*

[1]Department of Molecular Cellular and Developmental Biology, University of Michigan, Ann Arbor, United States; [2]Department of Developmental Biology, Washington University School of Medicine, St. Louis, United States; [3]Department of Cell and Developmental Biology, University of Michigan, Ann Arbor, United States; [4]Perelman School of Medicine, University of Pennsylvania, Philadelphia, United States; [5]Department of Neurology, University of Michigan School of Medicine, Ann Arbor, United States

*For correspondence: collinca@ umich.edu

**Competing interests:** The authors declare that no competing interests exist.

**Abstract** A broadly known method to stimulate the growth potential of axons is to elevate intracellular levels of cAMP, however the cellular pathway(s) that mediate this are not known. Here we identify the Dual Leucine-zipper Kinase (DLK, Wnd in *Drosophila*) as a critical target and effector of cAMP in injured axons. DLK/Wnd is thought to function as an injury 'sensor', as it becomes activated after axonal damage. Our findings in both *Drosophila* and mammalian neurons indicate that the cAMP effector kinase PKA is a conserved and direct upstream activator of Wnd/DLK. PKA is required for the induction of Wnd signaling in injured axons, and DLK is essential for the regenerative effects of cAMP in mammalian DRG neurons. These findings link two important mediators of responses to axonal injury, DLK/Wnd and cAMP/PKA, into a unified and evolutionarily conserved molecular pathway for stimulating the regenerative potential of injured axons.

## Introduction

Repair of lost axonal connections generally fails to occur after neuronal injury in the adult mammalian central nervous system (CNS). This failure is not only a reflection of the growth inhibitory nature of CNS tissue (*Fawcett et al., 2012*; *Filbin, 2003*; *Silver et al., 2015*), but also due to the lack of intrinsic capacity for neurons in the adult CNS to grow axons (*Liu et al., 2011*; *Sun and He, 2010*). However, landmark studies by Richardson and Issa have suggested that neurons indeed possess an innate ability to regenerate their axons in the adult mammalian CNS, and that this ability can be unlocked by a 'conditioning lesion' (*Richardson and Issa, 1984*). In adult DRG neurons, an injury to peripherally projecting axons, i.e. a compression injury to the sciatic nerve, unleashes growth programs within the DRG and allows for regeneration of its centrally projecting axons in the spinal cord (*Neumann and Woolf, 1999*; *Richardson and Issa, 1984*). This growth can be induced by a lesion in the peripheral nervous system (PNS) even after the CNS lesion has occurred (*Ylera et al., 2009*), hence it is of great interest from a therapeutic perspective to understand the molecular mechanisms that allow for the unlocking of such regenerative potential.

Previous studies have discovered that several signal transduction pathways are activated in DRG neurons upon a conditioning injury, including JAK-STAT3 (*Qiu et al., 2005*), ATF3 (*Fagoe et al.,*

**eLife digest** Adult mammals typically cannot repair damage to the nerve fibers in their brain or spinal cord. This is because these nerve cells cannot generally grow new nerve fibers. However this inability to regenerate nerve fibers is not set in stone. Instead, it can be unlocked by a second injury in nerves elsewhere in the body, the so-called "peripheral nervous system". This process relies on an enzyme called DLK, which becomes activated in damaged nerve fibers.

But how does DLK 'sense' damage to nerve fibers? Injuring the peripheral nervous system causes the levels of a molecule called cAMP to increase in the damaged nerve cells, and the elevated cAMP levels stimulate the nerve fibers to regenerate. However, it was not known if cAMP activates DLK, or if the two act independently of each other.

By looking at the regeneration of damaged nerve fibers in fruit fly larvae, Hao et al. now show that the cAMP and DLK signaling pathways are clearly linked. Further experiments with nerve cells from mice and human cells revealed more detail about this link. Together the results showed that another enzyme called PKA activates DLK directly when cAMP levels are high. These findings reveal a unified pathway that is the key to unlocking the regenerative potential of injured nerve fibers, which has been conserved for hundreds of millions of years of evolution. Further work could now ask if the DLK enzyme is involved in the other known roles of cAMP signaling in nerve cells; or if cAMP and PKA activate DLK in other forms of nerve damage, including injuries where nerve fibers normally fail to regenerate.

2015; *Hollis and Zou, 2012*), Smad1 (*Zou et al., 2009*), Activin (*Omura et al., 2015*), HIF-1alpha (*Cho et al., 2015*) and cAMP (*Qiu et al., 2002*; *Neumann et al., 2002*; *Cai et al., 1999*). Impressively, ectopic elevation of cAMP alone is sufficient to strongly enhance regeneration (*Xiao et al., 2015*; *Qiu et al., 2002*; *Neumann et al., 2002*). Since this second messenger is commonly modulated by growth signals and neuronal activity, cAMP modulation has been suggested as a potential therapeutic inroad to stimulate the regenerative potential of neurons (*Xiao et al., 2015*). However, the downstream pathways that are engaged by this broadly utilized second messenger to actually promote axonal regeneration are not known. Much attention has focused upon the cAMP-responsive element binding protein (CREB), since constitutive activation of CREB is sufficient to stimulate axonal regeneration in the presence of CNS myelin *in vivo* (*Gao et al., 2004*). However, more recent studies indicate that endogenous CREB is not required for cAMP elicited axonal regeneration *in vitro* (*Ma et al., 2014*). Hence it remains elusive how cAMP elevation activates axonal regrowth programs in neurons.

A recent study has identified an essential role for the dual zipper-bearing kinase DLK in the pro-regenerative effect of a conditioning lesion in adult DRG neurons (*Shin et al., 2012*). Similarly, the *Drosophila* homologue Wallenda (Wnd), mediates protective effects of a conditioning lesion in *Drosophila* motoneurons (*Brace and DiAntonio, 2016*; *Xiong and Collins, 2012*). This conserved axonal mitogen activated kinase kinase kinase (MAPKKK) is thought to function as a sensor of axonal damage, and therefore should become activated upon conditioning injury. In support of this, Wnd/DLK is transported in axons (*Xiong et al., 2010*) and is required acutely in injured axons for the generation of signals that are retrogradely transported to the cell body (*Xiong et al., 2010*; *Shin et al., 2012*). DLK/Wnd is required for axonal regeneration in many types of neurons, including motoneurons in mammals, flies and worms, and CNS neurons where regeneration is ectopically induced by PTEN mutations (*Yan et al., 2009*; *Hammarlund et al., 2009*; *Xiong et al., 2010*; *Shin et al., 2012*; *Watkins et al., 2013*). Conversely, in mammalian CNS neurons that do not regenerate (eg. retinal ganglion cells, RGCs), DLK activation after injury mediates cell death (*Welsbie et al., 2013*; *Watkins et al., 2013*).

Collectively, these findings support the model that a conserved function of the Wnd/DLK kinase is to 'sense' axonal damage. Through a yet unknown mechanism, axonal damage leads to activation of Wnd/DLK's kinase function. Once activated, downstream signaling mediates both beneficial and deleterious outcomes in neurons, depending upon the context. The high stakes outcomes of regeneration or death, combined with additional findings that DLK mediates cell death in models for nerve

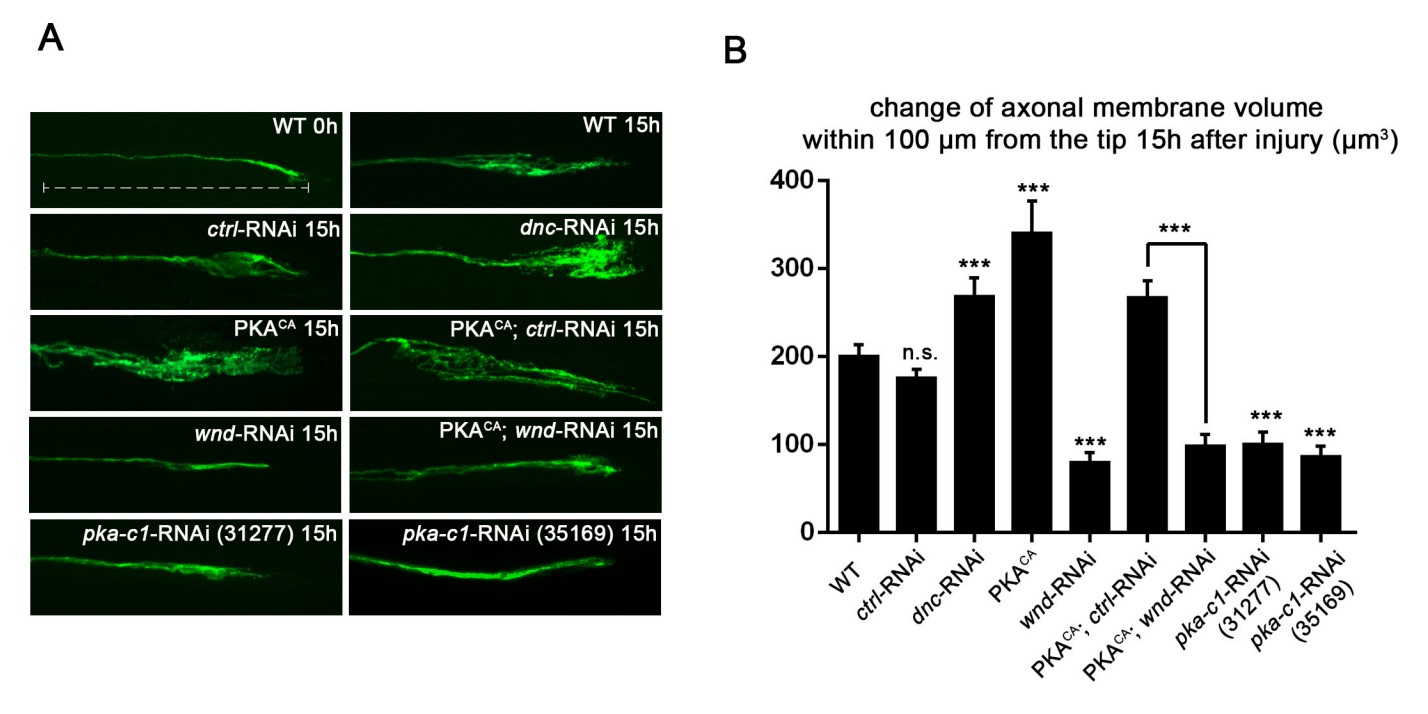

**Figure 1.** PKA stimulates and is required for axonal regeneration in *Drosophila* motoneurons. Single motoneuron axons were labeled by expression of UAS-mCD8-GFP using the m12-Gal4 driver and imaged either 0 hr or 15 hr after nerve crush injury. Representative images are shown in (A), while (B) shows quantification of the increased volume in axonal membrane, which is measured within 100 µm of the proximal axon tip, indicated in dotted line. Genotypes used in (A): wild type (WT) (;;m12-Gal4, UAS-mCD8GFP/+); *control*-RNAi (UAS-*dcr2*;; UAS-*moody*-RNAi/m12-Gal4, UAS-mCD8GFP); *dnc*-RNAi (UAS-*dcr2*;; UAS-*dnc*-RNAi/m12-Gal4,UAS-mCD8GFP); *wnd*-RNAi (UAS-*dcr2*;; UAS-*wnd*-RNAi/m12-Gal4,UAS-mCD8GFP); PKA$^{CA}$ (; UAS-PKA$^{CA}$/+; UAS-mCD8GFP/+); PKA$^{CA}$,*control*-RNAi (UAS-*dcr2*; UAS-PKA$^{CA}$/+; UAS-*moody*-RNAi (VDRC 100674)/m12-Gal4, UAS-mCD8GFP); PKA$^{CA}$,*wnd*-RNAi (UAS-*dcr2*; UAS-PKA$^{CA}$/+; UAS-*wnd*-RNAi/m12-Gal4, UAS-mCD8GFP); *pka-c1*-RNAi (UAS-*dcr2*; UAS-*pka-c1*-RNAi/+; m12-Gal4, UAS-mCD8GFP [using two different lines, Bloomington 31277 and 35169]). All data are represented as mean ± SEM; At least 10 animals (≥50 axons) are examined per genotype; ***p<0.001; 'n.s.' indicates non-significant; scale bar, 100 µm.

growth factor withdrawal (*Huntwork-Rodriguez et al., 2013*; *Ghosh et al., 2011*), glaucoma (*Welsbie et al., 2013*), MPTP toxicity (*Mathiasen et al., 2004*) and excitotoxicity (*Pozniak et al., 2013*), have inspired much interest in understanding the unknown pathways that lead to the activation of DLK/Wnd in injured axons.

Here we identify a direct upstream activator of DLK/Wnd in injured axons, in the form of the cAMP effector kinase PKA. We find that PKA phosphorylates evolutionarily conserved serines within the activation loop of DLK, which is sufficient to activate DLK independently of its downstream signaling mechanisms. In addition, our functional studies in both *Drosophila* motoneurons and adult mammalian DRG neurons indicate that the ability of cAMP and PKA to promote axonal regeneration depends entirely upon the ability of PKA to activate the DLK/Wnd kinase. These findings present a unified and evolutionarily conserved molecular pathway, from cAMP to PKA to DLK, which plays a central role in stimulating the ability of injured axons to regenerate.

## Results

### PKA regulates axonal regeneration via Wnd

Previous studies in mammalian and *C. elegans* neurons suggest that cAMP signaling stimulates regenerative axonal growth (*Qiu et al., 2002*; *Neumann et al., 2002*; *Cai et al., 1999*; *Ghosh-Roy et al., 2010*). To study this axon regeneration pathway in *Drosophila*, we used previously developed axon injury assay in third instar larvae (*Xiong et al., 2010*), and found that knockdown of phosphodiesterase *dunce* (*dnc*) or activation of PKA by overexpression of the catalytic subunit (PKA$^{CA}$)

(*Li et al., 1995*) led to an enhanced growth response of *Drosophila* motoneuron axons after nerve crush injury (*Figure 1A*). The new axonal growth from the injured proximal stump generally assumes a highly branched shape, characterized by a network of small branches and a general thickening of the axon diameter. To assess the injury response, we quantified the total membrane volume within 100 µm of the axonal tip (indicated by the dash line in *Figure 1A*). In control animals, this total volume increases 3 fold, from 68.5 $\mu m^3$ to 200 $\mu m^3$ 15 hr after injury. PKA activation led to a 1.5 fold increase in this volume compared to control (WT) axons (*Figure 1B*). The enhanced sprouting response stimulated by PKA was lost when DLK/Wnd function was inhibited by co-expression of RNAi targeting Wnd (but not a control RNAi) (*Figure 1A and B*). These observations are consistent with the previous finding in *C.elegans* that DLK is required for the regeneration that is induced by cAMP signaling (*Ghosh-Roy et al., 2010*). In addition, PKA alone is required for *Drosophila* motoneurons to initiate regenerative sprouting, as RNAi-mediated knockdown of the PKA catalytic subunit inhibited the sprouting response by 50% compared to control axons (*Figure 1A and B*). cAMP and PKA therefore play an influential role in the regenerative capacity of *Drosophila* motoneuron axons.

## PKA modulates the levels of Wnd and downstream signaling in *Drosophila* neurons

While the above and previously described genetic interactions (*Ghosh-Roy et al., 2010*) suggest a relationship, whether Wnd/DLK functions downstream of PKA or in a parallel pathway cannot be discerned from genetic epistasis alone. To probe the relationship between PKA and Wnd, we first utilized previously established tools in *Drosophila* for monitoring the activation of Wnd and downstream nuclear signaling. Wnd signaling induces expression of the c-Jun N-terminal Kinase (JNK) phosphatase puckered (*puc*), which can be measured as lacZ expression using fly lines that contain the *puc*-lacZ enhancer trap reporter (*Xiong et al., 2010*). *Puc*-lacZ is expressed at low levels in uninjured motoneurons, however it is induced by axonal injury in a manner that requires both Wnd and JNK kinase function (*Xiong et al., 2010*). We found expression of either *dnc*-RNAi or PKA[CA] induced the expression of *puc*-lacZ in motoneurons (*Figure 2A*). This induction is Wnd dependent, as RNAi knockdown of Wnd (but not a control RNAi) rescued the *puc*-lacZ elevation (*Figure 2A*).

Previous studies in multiple organisms suggest that Wnd/DLK is highly regulated at the level of protein turnover and increased levels of DLK correlate with the activation of downstream signaling (*Xiong et al., 2010*; *Huntwork-Rodriguez et al., 2013*; *Welsbie et al., 2013*; *Collins et al., 2006*; *Nakata et al., 2005*; *Nihalani et al., 2000*; *Hammarlund et al., 2009*). We therefore tested whether PKA activation altered Wnd levels. The total levels of endogenous Wnd within 2nd instar larval brains were significantly elevated (by 75%) when PKA[CA] was expressed in neurons (*Figure 2B*).

To test whether the change of Wnd level is due to a posttranscriptional mechanism, we used the Gal4/UAS system to ectopically express a GFP tagged Wnd transgene in *Drosophila* motoneurons. Since overexpression of WT Wnd can cause lethality, we expressed an inactive (kinase dead) version (GFP-Wnd[KD]) that contains a point mutation in the kinase domain (*Collins et al., 2006*). We found that expression of PKA[CA] induced a 12-fold increase in the levels of GFP-Wnd[KD] in motoneuron axons (*Figure 2C*). In contrast, GFP-Wnd[KD] was not significantly altered in cell bodies (*Figure 2C*).

The increase in axonal GFP-Wnd[KD] when PKA is induced has remarkable similarity to what occurs in axons after injury (*Xiong et al., 2010*; *Huntwork-Rodriguez et al., 2013*). We therefore tested whether PKA is required for the induction of Wnd in proximal axons after nerve crush injury. In control (WT) motoneurons, a significant increase in the mean intensity of the GFP-Wnd[KD] was observed 7 hr after injury (*Figure 2D*). This increase is abolished by co-expression of RNAi targeting PKA-C1, but not a control RNAi (*Figure 2D*). These observations suggest that PKA is required for the activation of Wnd signaling and the induction of Wnd protein levels downstream of axonal injury.

## PKA activates DLK through phosphorylation of its activation loop

To test whether the regulation of DLK by PKA is conserved in mammalian neurons, we examined endogenous DLK protein in cultured embryonic rat cortical neurons. Treatment with forskolin, which activates PKA via elevation of cAMP, led to a 2-fold increase in the level of endogenous DLK protein (*Figure 3A*). This effect of cAMP elevation requires PKA, since the increase in DLK levels was

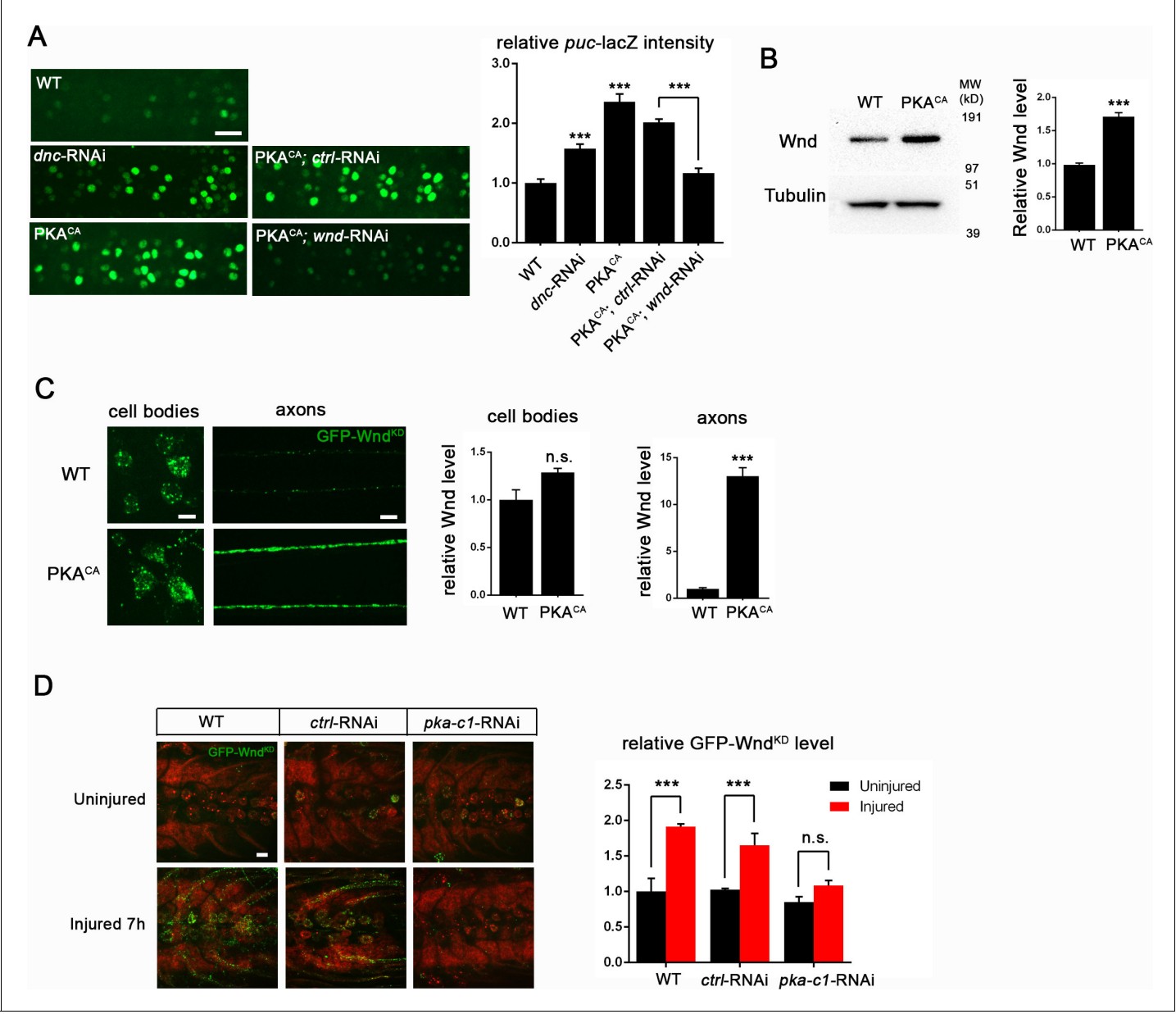

**Figure 2.** PKA modulates the levels of Wnd protein and downstream signaling in *Drosophila* neurons. (**A**) The *puc*-lacZ transcriptional reporter for Wnd/JNK signaling indicates that activated PKA stimulates Wnd signaling. A pan-neuronal driver (BG380-Gal4) is used to express UAS-*dnc*-RNAi, UAS-PKA^CA or UAS-PKA^CA together with UAS-*wnd*-RNAi or a *control*-RNAi. Example images are shown of cell bodies in the dorsal midline of the ventral nerve cord; (all but two of these neurons are motoneurons). Quantification (described in methods) was carried out for 10 animals per genotype. (**B**) Endogenous Wnd protein levels are increased in PKA^CA expressing neurons. Ventral nerve cords were dissected from third instar larvae (BG380-Gal4 [WT control] and BG380-Gal4; UAS-PKA^CA/+) and processed for Western blotting with anti-Wnd and anti-tubulin antibodies. The quantification shows Wnd/tubulin ratios (normalized to WT control) averaged from 3 independent experiments (25 nerve cords per experiment). (**C**) PKA increases DLK levels via a posttranscriptional mechanism. GFP-tagged kinase dead Wnd (GFP-Wnd^KD) was ectopically expressed using m12-Gal4 driver (WT control) or co-expressed with UAS-PKA^CA and imaged directly after fixation. Example images and quantification of GFP-Wnd^KD intensity in cell bodies and axons within segmental nerves. n>10 animals for each condition. (**D**) PKA-C1 is required for induction of Wnd protein after axonal injury. Example images and quantification of GFP-Wnd^KD in nerve cords and segmental nerves before and after (8 hr) injury. UAS-GFP-Wnd^KD was expressed in motoneurons by OK6-Gal4. WT or together with UAS-*pka-c1*-RNAi or UAS-*moody*-RNAi (control) and imaged similarly to *Figure 2C*. The quantification method for GFP intensity is described in materials and methods. n>10 animals for each condition. All data are represented as mean ± SEM; ***p<0.001, **p<0.01, *p<0.05, 'n.s.' indicates non-significant; scale bars, 10 µm.

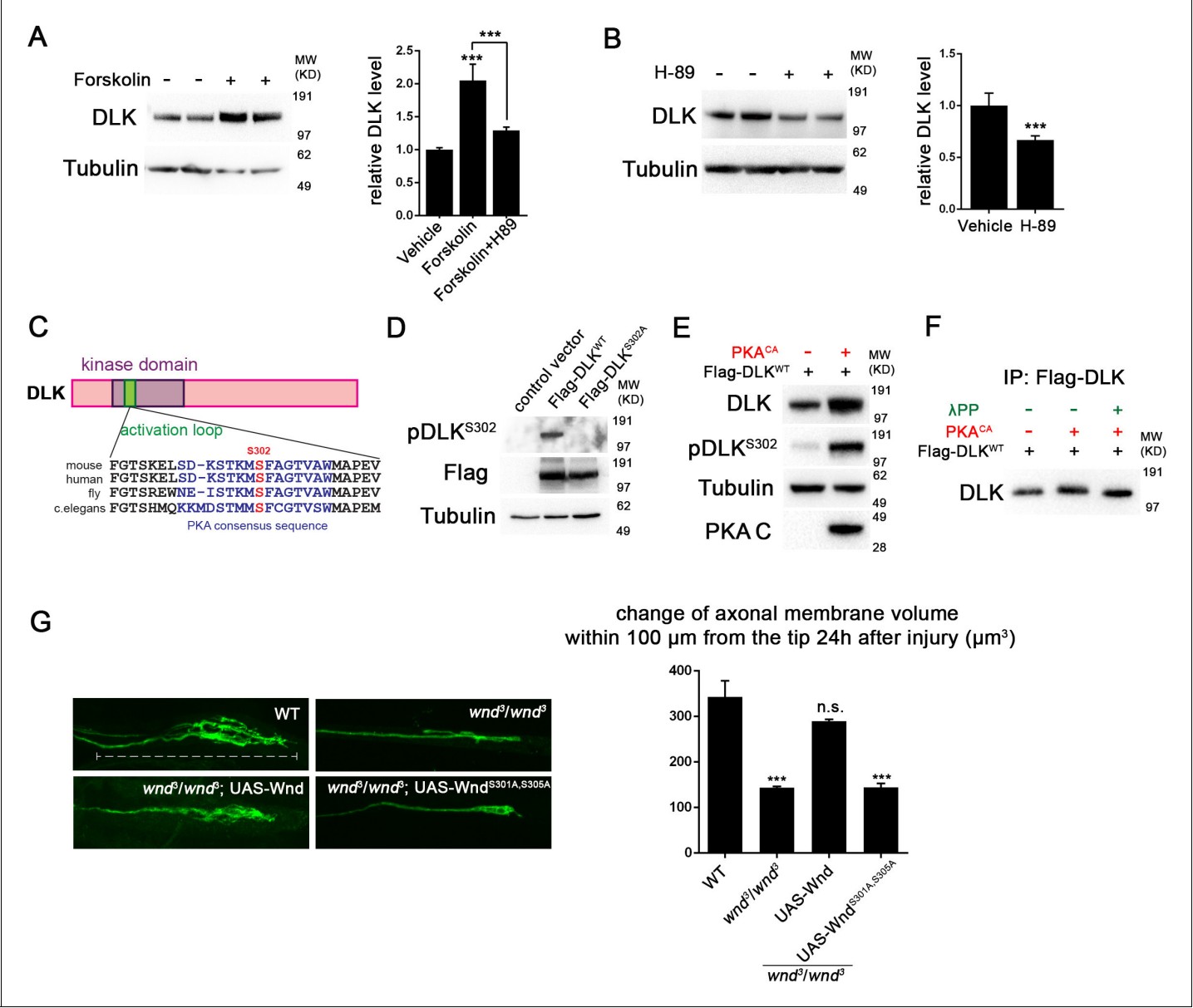

**Figure 3.** PKA activates DLK via phosphorylation of its activation loop. (**A-B**) Changes in endogenous DLK abundance in response to treatment with forskolin (30 μM) (**A**) or the PKA inhibitor H-89 (5 μM) (**B**) for 6 hr in cultured rat embryonic cortical neurons. Quantification shows relative DLK/Tubulin levels in western blots. (**C**) Alignment of activation loop sequences in different species. (**D**) The anti-pDLK$^{S302}$ antibodies recognize transfected Flag-DLK$^{WT}$, but not activation loop mutation Flag-DLK$^{S301A,S305A}$. Both proteins were transiently expressed in HEK293 cells. Western blots were probed with anti-pDLK$^{S302}$ antibody, anti-Flag antibody to detect the total DLK expression levels, and anti-Tubulin (which remains similar in all manipulations) for normalization. (**E**) PKA$^{CA}$ stimulates phosphorylation of DLK S302 in HEK293 cells. HEK293 cells were co-transfected with Flag-tagged DLK$^{WT}$ and an empty control plasmid or PKA$^{CA}$. Cell lysates were probed with anti-DLK antibody, anti-pDLK$^{S302}$ antibody, anti-PKA C antibody and anti-tubulin antibody. (**F**) PKA$^{CA}$ stimulates an increase in DLK molecular weight. Flag-DLK protein was immunoprecipitated from HEK293 cells co-transfected with DLK and either Flag-tagged DLK$^{WT}$ and an empty control plasmid or PKA$^{CA}$. The immunoprecipitated Flag-DLK was then incubated with either glycerol (control) or lambda protein phosphatase (λPP). PKA$^{CA}$ induced a upward shift in DLK molecular weight, which was lost upon phosphatase treatment. (**G**) The activation loop is required for axonal regeneration in *Drosophila* neurons. Single axons in *Drosophila* third instar larva are labeled by mCD8RFP using eve-Gal4 driver. 24 hr after injury, these neurons in animals heterozygous for *wnd* (*wnd$^3$*/+) show robust axonal sprouting. However, sprouting fails to occur in *wnd$^3$/wnd$^3$* animals. Expression of UAS-Wnd (WT) can restore axonal regeneration in *wnd* mutant background (UAS-Wnd, *wnd$^3$; wnd$^3$*, eve-Gal4, UAS-mCD8RFP). However, expression of activation loop mutant UAS-Wnd$^{S301A,S305A}$ failed to rescue the sprouting defect in *wnd* mutant animals (UAS-Wnd$^{S301A,S305A}$, *wnd$^3$; wnd$^3$*, eve-Gal4,m12-mCD8RFP). Quantification of the volume of axonal membrane within 100 μm of the distal ending of the proximal stump. n> 50 axons for each genotype. Data are presented as mean ± SEM for 3 independent experiments; ***p<0.001; scale bar, 100 μm.

*Figure 3 continued on next page*

*Figure 3 continued*

The following figure supplements are available for figure 3:

**Figure supplement 1.** PKA can directly phosphorylate DLK at S302.
**Figure supplement 2.** DLK activation by PKA does not require TORC1.

abolished by co-treatment with the PKA inhibitor H-89 (*Figure 3A*). In contrast, treatment with H-89 alone led to a significant reduction in the DLK levels (*Figure 3B*). Similar results were observed in HEK293 cells co-transfected with Flag-tagged DLK and either a control empty plasmid or PKA$^{CA}$. PKA$^{CA}$ induces an approximately two-fold increase in DLK protein levels (*Figure 3E* and *4B*). PKA$^{CA}$ also stimulates a phosphatase-sensitive increase in DLK molecular weight, which is most visible when equal amounts of DLK protein are compared (*Figure 3F*).

Although the mechanism of DLK activation is unknown, phosphorylation of the activation loop is important for activation of other kinases in the mixed lineage kinase family that DLK belongs to (*Durkin et al., 2004*; *Leung and Lassam, 2001*). Notably, the activation loop contains a conserved consensus sequence for PKA (*Figure 3C*), and a recent study has demonstrated that the predicted phosphorylation site S302 is required for DLK to activate downstream kinases (*Huntwork-Rodriguez et al., 2013*).

To test whether PKA stimulates phosphorylation of DLK's activation loop (S302), we have generated phospho-specific antibodies against a phosphorylated peptide corresponding to activation loop of mouse DLK (KELSDKpSTKMpSFAGTV). The phospho-DLK antibodies detected a strong band in HEK293 cells that overexpress WT mouse DLK, but show no reactivity for mutant DLK$^{S302A}$ (*Figure 3D*). Remarkably, expression of PKA$^{CA}$ in HEK293 cells induces a dramatic increase of phospho-S302 DLK (*Figure 3E*), even when normalized to the levels of total DLK (*Figure 4B*). Similar results were observed when cells were treated with forskolin for 3 hr (*Figure 4A*).

Since the activation loop contains a conserved consensus sequence for PKA substrates, it should be capable of phosphorylating this site in DLK directly. Indeed, we found that purified PKA can strongly stimulate pS302 reactivity upon purified Flag-DLK *in vitro* (*Figure 3—figure supplement 1*).

A recent study in *Drosophila* has suggested that TORC1 may activate and phosphorylate Wnd (*Wong et al., 2015*), so we considered whether TORC1 plays a role in the activation of DLK by PKA. We used both torin1 and rapamycin to inhibit TORC1 in the presence or absence of PKA in HEK293 cells. However, in both cases we observed no effect upon DLK and pDLK levels (*Figure 3—figure supplement 2*). PKA therefore stimulates phosphorylation of DLK's activation loop independently of TORC1 function.

To confirm that S302 is a critical site for Wnd/DLK function in axonal regeneration, we conducted a rescue experiment in *Drosophila* motoneurons based on the requirement of Wnd for axonal sprouting after injury. We generated UAS-Wnd$^{S301A,S305A}$ transgenic flies expressing Wnd with mutations in two serines analogous to S298 and S302 in the activation loop of DLK. As shown in *Figure 3G*, all axons in *wnd* mutants fail to initiate a sprouting response, which can be rescued by co-expression of WT Wnd, but not Wnd$^{S301A,S305A}$. In addition, overexpression of Wnd$^{S301A,S305A}$ does not give rise to any of the previous described gain-of-function phenotypes similar to WT Wnd, suggesting that S302 is indeed required for DLK function.

## PKA promotes DLK stability independently of downstream signaling

A previous study has described a positive feedback loop for DLK stabilization that involves the action of DLK's downstream effector JNK (*Huntwork-Rodriguez et al., 2013*). JNK activation stimulates phosphorylation of DLK at sites outside of its activation loop (T43 and S533) and changes DLK's sensitivity to degradation via the ubiquitin proteasome system (UPS) (*Huntwork-Rodriguez et al., 2013*). This increase in protein stability leads to an increase in total levels of DLK. Since PKA stimulates an increase in ectopically expressed DLK, it most likely increases protein stability. We therefore tested whether this increased stability involves the previously described JNK-dependent feedback mechanism. If this is the case, the effects of PKA and forskolin should depend on the function of JNK. As shown in *Figure 4A and B*, treatment with JNK inhibitor VIII led to a 30% decrease in total

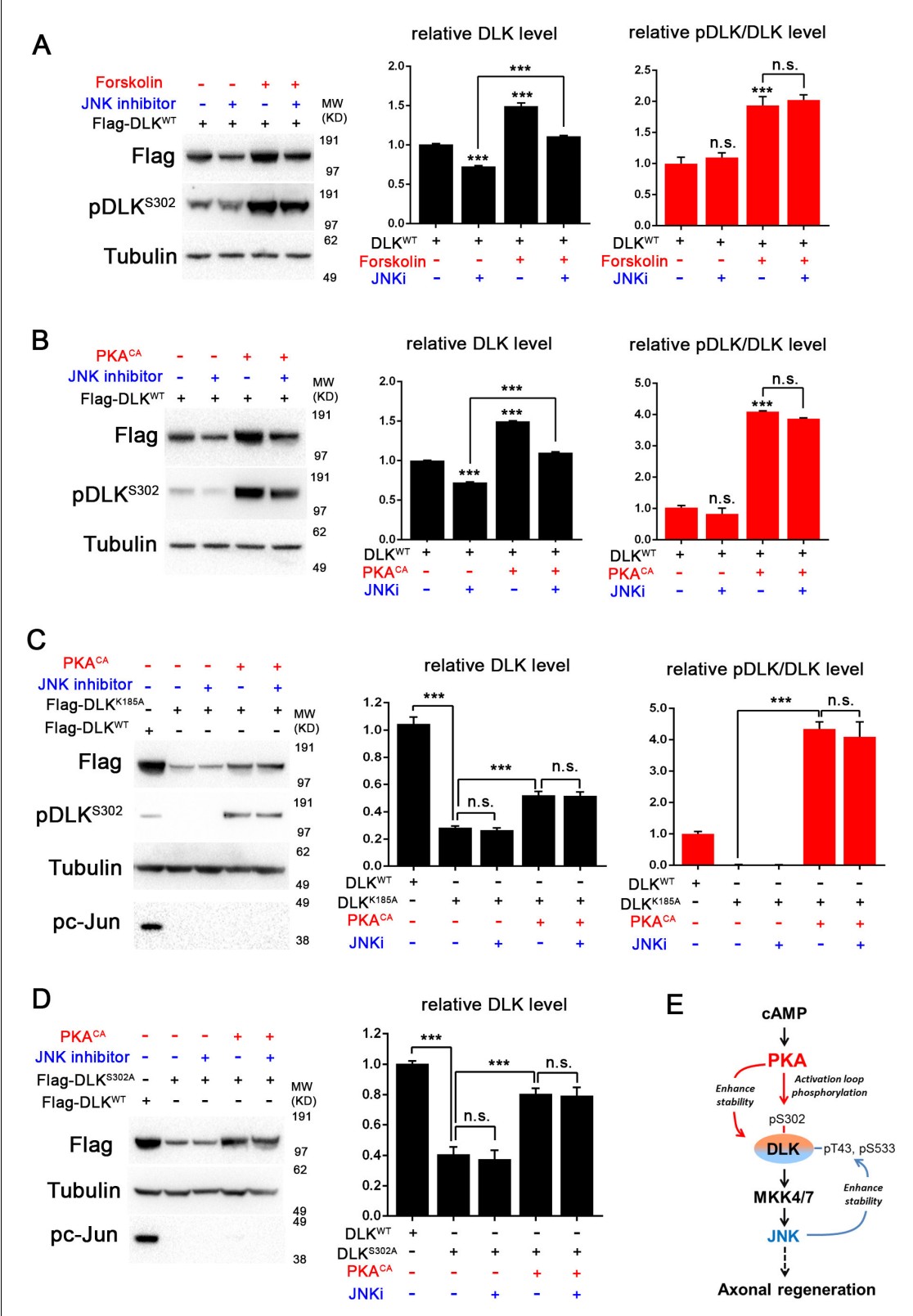

**Figure 4.** PKA promotes the stability of DLK independently of DLK downstream signaling. (A-B) Activation of PKA promotes DLK stability independently of JNK. HEK293 cells were transiently transfected with Flag-DLK^WT, and (A) treated with forskolin (6 hr, 30 μM) or (B) co-transfected with either PKA^CA or empty vector (control). In both cases, co-treatment with JNK inhibitor VIII (10 μM, 6 hr) led to a decrease in total Flag-DLK levels. However, both forskolin and PKA^CA induced an increase in DLK levels even in the presence of JNK inhibitor. Quantification shows average total DLK/

*Figure 4 continued on next page*

*Figure 4 continued*

Tubulin intensities and average pDLK$^{S302}$/total DLK ratios (where total DLK is detected using anti-Flag antibody). All data are represented as mean ± SEM; quantifications of relative intensity from Western Blots were averaged from 3 independent experiments; ***p<0.001, **p<0.01, *p<0.05, 'n.s.' indicates non-significant. (C-D) Activation of PKA increases the stability of kinase dead DLK mutants, DLK$^{K185A}$ (C) and DLK$^{S302A}$ (D). HEK293 cells were transiently transfected with Flag-DLK$^{K185A}$ or Flag-DLK$^{S302A}$ together with PKA$^{CA}$ or empty plasmid. Treatment JNK inhibitor VIII (10 μM) for 6 hr had no effect upon the PKA$^{CA}$ induced levels of DLK$^{K185A}$ and DLK$^{S302A}$ mutant protein. Quantifications are similar to *Figure 4A–B*. Western bands intensity were averaged from 4 independent experiments; data are shown as mean ± SEM; ***p<0.001, **p<0.01, *p<0.05, 'n.s.' indicates non-significant. (E) Proposed model for the activation and stabilization of DLK by cAMP and PKA. cAMP elevation and PKA activation leads to the phosphorylation of S302 on DLK, which activates its kinase activity. Indicated in the blue arrow, downstream signaling via JNK leads to enhanced DLK stability and phosphorylation of DLK at other sites (*Huntwork-Rodriguez et al., 2013*). PKA also enhances DLK's stability via an additional mechanism that is independent of S302 (red arrow).

The following figure supplement is available for figure 4:

**Figure supplement 1.** Summary of predicted PKA phosphorylation sites on DLK/Wnd in different species.

---

DLK level, as expected for JNK's previously demonstrated role in promoting DLK stability. However, even in the presence of JNK inhibitor, treatment with forskolin (*Figure 4A*) or transfection with PKA$^{CA}$ (*Figure 4B*) increases DLK level by 50%. Moreover, treatment with JNK inhibitor had very little effect upon the fraction of total DLK that is phosphorylated at S302 (*Figure 4A and B*). These results suggest that PKA stimulates DLK phosphorylation and stabilization independently of downstream JNK activation.

Previous biochemical studies suggest that DLK's activation mechanism is associated with dimerization and autophosphorylation (*Nihalani et al., 2000*; *Mata et al., 1996*; *Merritt et al., 1999*). We therefore further considered whether PKA could function either upstream or downstream of DLK's own ability to function as a kinase. To test this, we utilized a kinase-dead version of DLK, DLK$^{K185A}$, which is unable to activate downstream signaling or undergo autophosphorylation (*Nihalani et al., 2000*; *Mata et al., 1996*; *Merritt et al., 1999*). Consistent with the previously described feedback mechanism (*Huntwork-Rodriguez et al., 2013*), the DLK$^{K185A}$ mutant protein was less stable, and addition of JNK inhibitor had no further effect upon the levels of kinase dead DLK protein (*Figure 4C*). However, PKA$^{CA}$ stimulated a strong increase of DLK$^{K185A}$ levels and phosphorylation at S302 for DLK$^{K185A}$. The ability of PKA to increase DLK protein stability and activation loop phosphorylation independently of DLK's own signaling abilities places PKA firmly upstream of DLK, as an upstream regulator/activator.

Since PKA promotes DLK stability and directly phosphorylates S302, we wondered whether PKA stabilizes DLK by phosphorylation of S302. Previous work has shown that a decreased stability for DLK$^{S302A}$ mutant protein is linked to the fact that it is inactive for kinase activity, and is therefore unable to activate the downstream stabilization mechanism via JNK (*Huntwork-Rodriguez et al., 2013* and *Figure 4D*). Consistent with this, the reduced levels of DLK$^{S302A}$ protein are not further reduced in the presence of JNK inhibitors (*Figure 4D*). However surprisingly, co-transfection with PKA$^{CA}$ still caused an increase of the levels of DLK$^{S302A}$ protein (*Figure 4D*). We interpret that PKA regulates DLK via an additional mechanism, in conjunction with phosphorylation of the critical activation loop S302. This additional mechanism may involve other sites of phosphorylation, or other modes of regulation (discussed further below).

## PKA stimulates axonal regeneration via DLK in adult DRG neurons

Our finding that PKA activates DLK, taken together with previous findings that DLK promotes axonal regeneration in different neuronal cell types (*Shin et al., 2012*; *Xiong et al., 2010*; *Hammarlund et al., 2009*; *Yan et al., 2009*), led to the hypothesis that DLK is an important downstream mediator of cAMP-stimulated axon regeneration. To test this hypothesis, we employed a recently described replating assay for DRG neurons cultured from adult mice, which allows for a controlled and quantitative measure of the induction of axonal regeneration by *in vitro* manipulations such as forskolin treatment (*Frey et al., 2015*; *Valakh et al., 2015*). In this assay (depicted in *Figure 5A*), DRG neurons removed from adult mice were first cultured for 4–5 days, which allowed for the regenerative response activated by the dissection to subside. Neurons were then treated with forskolin for 24 hr, and then replated onto a fresh dish. The replating process removes all

existing neurites so that the neurites observed within the second culture period can be identified as new growth. As shown in *Figure 5B and E*, treatment with forskolin stimulates the regenerative response (*Frey et al., 2015*). In addition, co-treatment with H-89 abolished the effect of forskolin on neurite outgrowth (*Figure 5B–D*), suggesting that PKA is required.

To determine whether forskolin-induced neurite outgrowth requires DLK, we performed the same experiment in neurons from DLK (Map3k12) conditional knockout (KO) mice and littermate controls. The effects of forskolin on neurite outgrowth were abolished in DLK conditional knockout DRG neurons (*Figure 5E–G*). These findings suggest that DLK and its downstream signaling pathway(s) are important mediators of the pro-regenerative effects of cAMP elevation in neurons.

## Discussion

The Wnd/DLK kinase is likely to function as a sensor of axonal damage. Depending upon the context, its activation can lead to either axonal regeneration or cell death and degeneration (*Tedeschi and Bradke, 2013*). The factors that determine beneficial versus detrimental outcomes, along with the general cellular mechanisms that lead to the activation of DLK, are poorly understood. In this study, we found that an immediate upstream activator of DLK is the cAMP regulated kinase PKA. Elevation of cAMP signaling, which is activated by pro-regenerative manipulations such as a conditioning lesion, is the most widely known pathway for promoting axonal regeneration (*Hannila and Filbin, 2008*). We found that an essential component of this regenerative pathway is the activation of Wnd/DLK by PKA. These findings delineate an evolutionarily conserved mechanism for the activation of the Wnd/DLK kinase. Taken together with previous findings that Wnd/DLK is an essential regulator of responses induced by a conditioning injury (*Shin et al, 2012*; *Xiong and Collins 2012*), the activation of Wnd/DLK by PKA in injured axons presents a unified molecular pathway for activating a regenerative response to axonal damage.

In contrast to a merging of cAMP and DLK pathways, some other studies have suggested that these pathways may act separately (*Li et al., 2015*; *Chung et al., 2016*). A recent study has noted that in certain sensory neuron types in *C. elegans*, PKA *gain-of-function* mutations can induce axonal outgrowth even in *dlk* mutants (*Chung et al., 2016*). Hence, multiple pathways for axonal regeneration may be inducible by PKA. However DLK is strongly required for cAMP stimulated regeneration in other neuron types in *C. elegans* (*Ghosh-Roy et al., 2010*), and, importantly, in axonal regeneration induced by a conditioning lesion in the mammalian PNS (*Shin et al., 2012*). Our findings now indicate that DLK is an important molecular target and effector of cAMP-induced regeneration in mammalian neurons.

Previous biochemical studies indicate that DLK activation involves dimerization via its leucine zipper domains and autophosphorylation, at locations that are yet undefined (*Nihalani et al., 2000*; *Mata et al., 1996*). Because ectopic elevation of DLK/Wnd protein is sufficient to activate its downstream signaling (*Mata et al., 1996*; *Nihalani et al., 2000*; *Huntwork-Rodriguez et al., 2013*), and DLK is known to be highly regulated at the level of protein turnover (*Collins et al., 2006*; *Xiong et al., 2010*; *Huntwork-Rodriguez et al., 2013*; *Nakata et al., 2005*; *Hammarlund et al., 2009*), a plausible mechanism for its regulation is to hold its levels and/or its ability to dimerize in check (*Mata et al., 1996*; *Nihalani et al., 2000*). The existence of a direct upstream activator of the kinase was not previously implied, and has thus far been unknown. Here we found that PKA stimulates the phosphorylation of the activation loop of DLK independently of DLK's kinase activity, and also independently of downstream JNK signaling. This defines PKA as an upstream activator of DLK.

A previous study in *C. elegans* has described a mechanism through which transient elevation of intracellular $Ca^{2+}$ upon axonal injury leads to the activation of DLK-1 (*Yan and Jin, 2012*; *Cho et al., 2013*; *Ghosh-Roy et al., 2010*; *Spira et al., 2001*). In addition, earlier studies have implicated calmodulin-regulated calcineurin in the regulation of mammalian DLK (*Mata et al., 1996*). However, the hexapeptide that mediates activation by $Ca^{2+}$ in *C. elegans* is not present in mammalian or *Drosophila* DLK/Wnd, and mammalian DLK can be activated independently of $Ca^{2+}$ elevation by cytoskeletal destabilizing agents (*Valakh et al., 2015*). In contrast, the consensus PKA phosphorylation site in the activation loop of Wnd/DLK is conserved in all phyla (*Figure 3C*), suggesting this pathway as a central (although not necessarily exclusive) mechanism for regulating DLK.

We note that in conjunction with phosphorylation of DLK's essential activation loop, PKA enhances DLK's stability via an additional mechanism (*Figure 4E*), since the mutated protein DLK[S302A] can

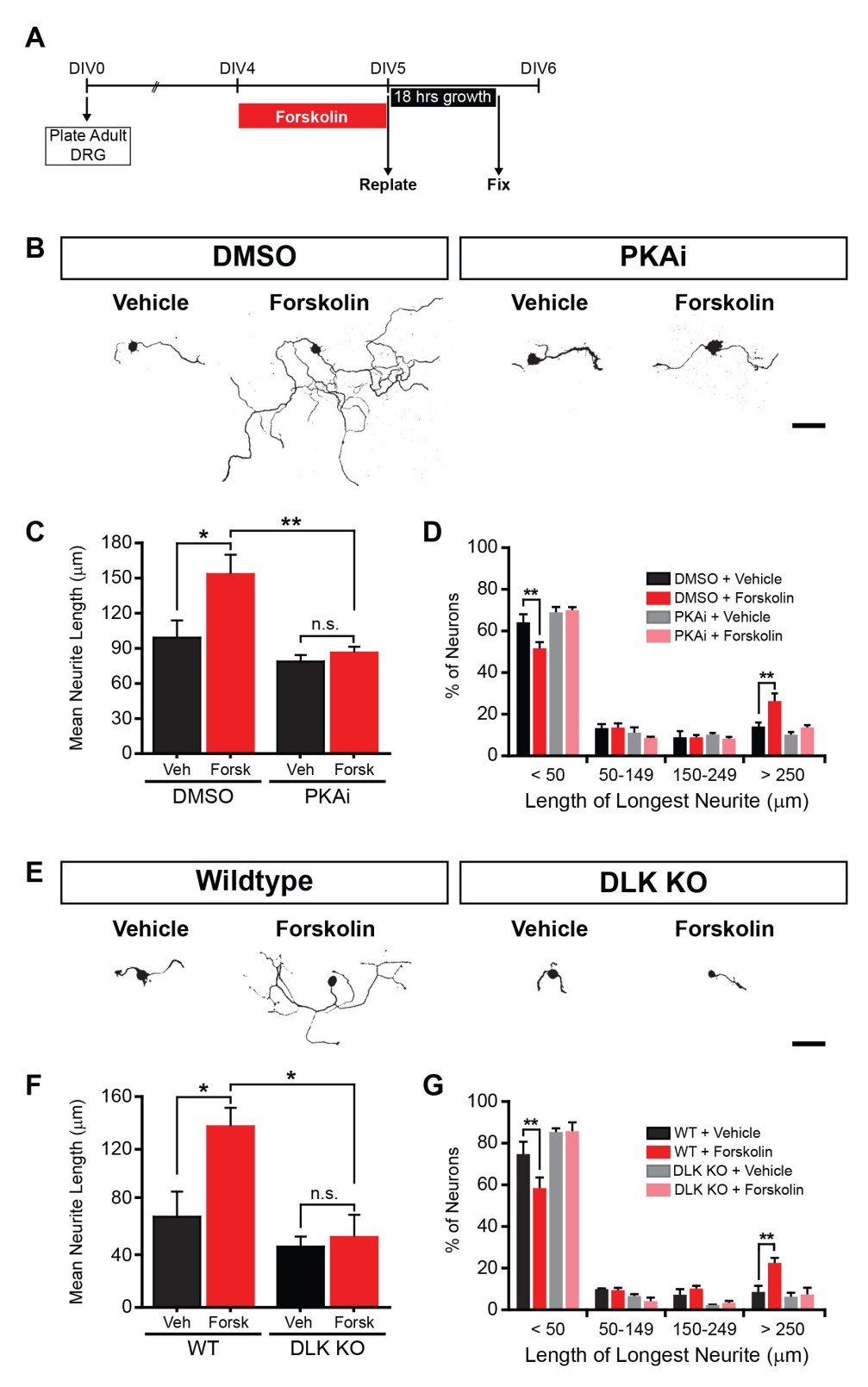

**Figure 5.** PKA stimulates axonal regeneration via DLK in adult DRG neurons. (**A-D**) Induction of regeneration by forskolin requires PKA. Experimental design (**A** and also see Materials and Methods). To demonstrate that forskolin-induced neurite outgrowth is mediated by PKA, we assessed whether PKA signaling was required using the PKA inhibitor H-89 (PKAi, 5 μM). Representative neurons are shown in (**B**). Neurite outgrowth was assessed by quantifying mean neurite length (**C**) and distribution of longest neurite (**D**). Data are mean ± SEM for 4 independent experiments. (**E-G**) Induction of

*Figure 5 continued on next page*

Figure 5 continued

regeneration by forskolin requires DLK. WT and DLK KO neurons were treated with DMSO or forskolin (30 µM) as described in (A). Representative neurons are shown in (E). Neurite outgrowth was assessed by quantifying mean neurite length (F) and distribution of the longest neurite (G). Data are mean ± SEM for 3 independent experiments. ***p<0.001, **p<0.01, *p<0.05, 'n.s.' indicates non-significant; scale bars, 100 µm.

The following source data is available for figure 5:

**Source data 1.** Measurements of the longest neurite length for 100 neurons after replating in each condition.

still be stabilized upon PKA activation. This mechanism may involve additional phosphorylation sites on DLK, and indeed, multiple PKA consensus sequences are observed in Wnd/DLK's sequence (*Figure 4—figure supplement 1*). However, it is also possible that PKA regulates DLK's stability via other mechanisms, such as previously described ubiquitination (*Nakata et al., 2005*; *Collins et al., 2006*), palmitoylation modification (*Holland et al., 2016*), changes in DLK's interacting proteins or subcellular localization. Such aspects of regulation could involve additional molecular targets of PKA, which would provide some potential for context specificity in DLK's regulation. While PKA has many cellular targets, its specificity can be highly regulated at a subcellular level by interactions with AKAP scaffolding proteins and local changes in cAMP (*Tasken and Aandahl, 2004*; *Wong and Scott, 2004*). It will be interesting to identify the additional players in cAMP and PKA regulation of DLK through future work.

It is remarkable that PKA stimulates a specific increase in Wnd levels in axons but not cell bodies (*Figure 2C*), and inhibition of PKA strongly inhibits the induction of Wnd protein after axonal injury (*Figure 2D*). Does PKA act locally in axons to stimulate DLK? Other studies have suggested that Wnd/DLK can regulate retrograde signaling pathways that originate in axons (*Xiong et al., 2010*; *Ghosh et al., 2011*; *Shin et al., 2012*; *Huntwork-Rodriguez et al., 2013*; *Watkins et al., 2013*; *Yan et al., 2009*; *Holland et al., 2016*). Intriguingly, the Hiw/Rpm-1/Phr1 ubiquitin ligase, which is previously known for its role in regulating Wnd/DLK's levels in axons (*Collins et al., 2006*; *Nakata et al., 2005*; *Lewcock et al., 2007*; *Babetto et al., 2013*), contains a RCC1-like domain that biochemically inhibits adenylate cyclase, and therefore may negatively regulate cAMP signaling (*Pierre et al., 2004*). Hiw/Rpm-1/Phr1 is previously known for its role in regulating synaptic arborization and growth via its regulation of Wnd/DLK (*Nakata et al., 2005*; *Collins et al., 2006*; *Wang et al., 2013*; *Wan et al., 2000*; *Shin and DiAntonio, 2011*). The addition of cAMP and PKA into this regulatory pathway suggests a mechanism that may be broadly utilized to orchestrate structural changes within presynaptic terminals. We propose that the regulation of DLK by PKA may be generally important for neuronal plasticity as well as responses to axonal damage.

## Materials and methods

### Fly genetics

The following fly strains were used in this study: Canton-S (WT), m12-Gal4 (*Ritzenthaler et al., 2000*), BG380-Gal4 (*Budnik et al., 1996*), OK6-Gal4 (*Aberle et al., 2002*), RRa(eve)-Gal4 (*Fujioka et al., 2003*), *puc*-lacZ$^{E69}$, *wnd*$^3$, UAS-Wnd, (*Collins et al., 2006*), UAS-GFP-Wnd$^{KD}$ (*Xiong et al., 2010*), UAS-PKA$^{CA}$ (*Li et al., 1995*). UAS-Wnd$^{S301A,S305A}$ flies were generated from pUAST-Wnd$^{S301A,S305A}$ plasmid for this study. UAS-*wnd*-RNAi (VDRC 13786) and UAS-*moody*-RNAi (VDRC 100674) were from the Vienna RNAi center (*Dietzl et al., 2007*). UAS-*pka-c1*-RNAis (31277, 35169) and UAS-*dnc*-RNAi (27250) were acquired from Bloomington stock center.

### Nerve crush assay and immunocytochemistry in *Drosophila*

Peripheral nerve crush assays in 3$^{rd}$ instar larvae were performed according to *Xiong et al. (2010)*. Briefly, the segmental nerves of third instar larvae were pinched and crushed by a fine No.5 forceps while the larvae were anesthetized with $CO_2$ gas. After injury, larvae were transferred to a grape plate and kept in 25°C incubator for specified time periods.

*Drosophila* third instar larva were dissected in ice-cold PBS and fixed in 4% paraformaldehyde for 25 min. Antibodies were used in PBS supplemented with 0.3% Triton and 5% normal goat serum. Anti-lacZ (40-1a, Developmental Studies Hybridoma Bank) was diluted 1:100. Anti-dsRed polyclonal

antibody (632495, Clontech) was diluted 1:1000. For secondary antibodies, A488- or Cy3-conjucated goat anti-mouse or rabbit (Invitrogen, Carlsbad, CA) were used at 1:1000.

## Imaging and quantification

Confocal images were collected on an Improvision spinning disk confocal system, consisting of a Yokagawa Nipkow CSU10 scanner, and a Hamamatsu C1900-50 EMCCD camera, mounted on a Zeiss Axio Observer with 40X (1.3NA) oil objectives. Similar settings were used for imaging of all compared genotypes and conditions. Volocity software (Perkin Elmer) was used for intensity measurements and quantification of all confocal data.

A quantification of the sprouting response in injured motoneuron axons was measurement of the change of total volume of regenerating axonal membrane, labeled by mCD8-GFP, within a 100 μm distance from the injured tip. Pixels within the most distal 100 μm of the injured proximal stump were selected based on mCD8-GFP intensity criteria of >3 standard deviation above the mean, and then summed to measure total membrane volume. *Figure 1* and *3F* report the change in average volume at 15 hr after injury compared to T=0 (immediately after injury).

*Puc*-lacZ expression was quantified by measurement of mean intensity for lacZ staining in the nuclei of motoneurons located along the dorsal midline (segments A3-A7) of the nerve cord of third instar larva. The mean intensity of GFP-Wnd$^{KD}$ within segmental nerves was quantified by measuring GFP intensity within 100 μm distance of each nerve at the site of exit from the ventral nerve cord.

## Cell culture

HEK293 cells were cultured in DMEM/F12 (Gibco) supplemented with 10% fetal bovine serum (Gibco) and 1% penicillin-streptomycin (Gibco). For transient transfection, 1 μg of given vectors were transfected into 3.5 cm dish using Lipofectmine 2000 (Invitrogen). 20 hr after transfection, cells were processed for Western blotting. Plasmids used for transfection were Flag-DLK (*Huntwork-Rodriguez et al., 2013*), Flag-DLK$^{S302A}$ (*Huntwork-Rodriguez et al., 2013*), Flag-DLK$^{K185A}$ (directly generated from Flag-DLK) and PKA$^{CA}$ (*Merrill et al., 2011*).

## Primary neuron culture and mouse model

Cortical neurons were dissected from E18 rat embryos. The cortex was digested by incubating with 0.5% trypsin-EDTA (Gibco) and DNAse I (Roche) at 37°C for 10 min. Following digestion, neurons were washed twice in DMEM medium (Gibco) containing 10% FBS before resuspension in neuronal growth media which containing neurobasal (Gibco), Glucose (Sigma), Glutamax (Gibco), penicillin-streptomycin and B27 supplement (Gibco). All the plates were coated with 100 μg/mL ploy-D-lysine (P7886, Sigma) for 2 hr. Neurons were then triturated and plated at a final concentration 400,000 cells/mL.

For adult DRG experiments, DRG neurons were collected from either CD1(Charles River), or Map3k12 F/F; Advillin-Cre-/- (WT), or Map3k12 F/F; Advillin-Cre+/- (DLK KO) mice. WT and DLK KO mice were age-sex matched. Neurons were prepared as previously described (*Frey et al., 2015*). Briefly, DRG were digested for 15 min at 37°C with 0.35 mg/mL liberase Blendzyme (Roche), 10 mg/mL bovine serum albumin (Sigma), and 0.6 mg/mL DNase (Sigma) followed by another 15 min digest at 37°C with 0.05% trypsin. DRG were tritterated in culture media (DMEM containing 10% FBS, 100 μg/mL penicillin and 100 μg/mL streptomycin) to dissociated cells. Cells were plated on PDL (10 mg/mL) and laminin (10 mg/mL) coated plates. On day *in vitro* (DIV)1, half of the media was removed and fresh media containing AraC (Sigma, 10 nM final) was added. Drug treatment and replating were performed as described previously (*Frey et al., 2015*). On DIV4, DMSO or forskolin (30 μM) were added to cells. 24 hr after drug application, drugs were washed out with DMEM. Neurons were lightly trypsinized with 0.025% trypsin for 5 min in the incubator (37°C, 5%CO$_2$). Trypsin was removed and fresh culture media was added to the cells which were then gently pipetted and transferred to culture slides. 18 hr after replating, neurons were fixed (4% PFA) and stained for βIII-tubulin (Covance, mouse anti-Tuj1, 1:500). At least 100 neurons were imaged per group using either Leica DFC310 FX or DFC7000T color fluorescence cameras and longest neurite was traced using NeuronJ plugin for ImageJ. For replating experiments with PKA inhibitor (H-89, 5 μM), vehicle or inhibitor were added at the same time as DMSO and forskolin.

## Immunochemistry, antibodies and chemicals

For detection of Wnd protein levels in larva nerve cords, the whole brains were carefully dissected from third instar larva. The two brain lobes were removed before they were frozen in liquid nitrogen and processed for western blotting (25 nerve cords per lane).

For western blots using HEK293 cells or cortical neurons, cells are lysed by incubating on ice for 10 min with RIPA buffer (BP-115, Boston BioProducts) supplemented with Complete protease inhibitor cocktail (Roche) and PhosSTOP phosphatase inhibitor (Roche). Protein concentrations were measured by BCA assay kit (Thermo Scientific).

Equal amount of protein samples were loaded on each lane of NuPAGE 4–12% Bis-Tris gels (Invitrogen) and subject to electrophoresis separation in MOPS buffer (Invitrogen). Blots were visualized with SuperSignal chemiluminescent substrate (Thermo Scientific) and exposure to either film or ChemiDoc (Bio-Rad). Bands intensities were determined using software ImageJ (NIH) using the gel analysis plug-in.

The following antibodies were used for Western blotting: anti-Wnd A3-1,2 at 1:700 (*Collins et al., 2006*); anti-DLK at 1:5000 (*Huntwork-Rodriguez et al., 2013*); anti-β-tubulin at 1:1000 (E7; Developmental Studies Hybridoma Bank); anti-Flag at 1:1000 (F1804, Sigma); anti-PKA C (catalytic subunit) antibody at 1:1000 (4782, Cell signaling); anti-phospho-cJun antibody at 1:1000 (3270, Cell signaling); anti-phospho-S6K antibody at 1:1000 (9234, Cell signaling) and anti-S6K antibody at 1:1000 (2708, Cell signaling).

Anti-phospho-DLK$^{S302}$ antibodies were raised by immunization of rabbits with the peptide KELSDKpSTKMpSFAGTV and affinity purified before use at dilution 1:100. While the antibodies were raised against a dually phosphorylated peptide (pS298, pS302), no difference in reactivity was noticed for DLK S298A mutants (data not shown).

Forskolin (F6886, Sigma) was applied to either HEK293 cells or cortical neurons at the final concentration of 30 μM for 6 hr. H-89 (B1427, Sigma) was used at the final concentration of 5 μM for cortical neurons for 6 hr. JNK inhibitor VIII (420135, EMD Millipore) was used at the final concentration of 10 μM on HEK293 cells for 6 hr. Torin1 (4247, Tocris) or Rapamycin (LC laboratories) was applied to the cells at 1 μM (final concentration) for 2 hr before harvest.

## Protein phosphatase assay

24 hr after transfection of the given constructs, HEK293 cells in a 6 cm dish were washed by ice-cold PBS and harvested in ice-cold RIPA buffer supplemented with EDTA-free protease inhibitor (Roche). The cell lysates were incubated on ice for 30 min and centrifuged at 14,000 rpm at 4°C for 10 min. The soluble lysate was incubated with 3 μg anti-Flag antibody pre-bound with 15 μl Dynabeads Protein G (Novex) for 1 hr at 4°C. After removal of supernatants the beads were washed 3 times with RIPA buffer, and then incubated with 1X PMP buffer (NEB), 1X $MnCl_2$ (NEB) and either 1000 U lambda protein phosphatase (NEB) or equal amount of glycerol (control) in RIPA buffer for 30 min at 30°C. Beads were then removed from the reaction buffer and Flag-DLK was eluted by boiling in SDS sample buffer for 10 min. Equal amount of samples were analyzed by western blot.

## *In vitro* PKA kinase assay

HEK293 cells transfected with Flag-DLK were washed by ice-cold PBS and harvested using ice-cold cell lysis buffer (50 mM Tris-HCl, pH7.5, 150 mM NaCl, 1 mM EDTA, 1% Triton X-100, Complete protease inhibitors (Roche) and 30 μM MG132 (Sigma)). Flag-tagged DLK was immunoprecipitated from HEK293 cell lysates using anti-Flag M2 antibody (Sigma) and Protein G Dynabeads (Novex). The DLK-bound beads were then washed with 3 times with lysis buffer and incubated with 2000 U lambda protein phosphatase (NEB) for 30 min at 30°C to remove all the phosphate groups. After incubation, the beads were washed with wash buffer containing cocktail phosphatase inhibitor (Roche) and split equally into two tubes containing kinase reaction buffer (50 mM Hepes, pH7.2, 10 mM $MgCl_2$, 0.01% Triton X-100, 2 mM DTT, and 30 μM ATP). 10,000 U recombinant human full length PKA catalytic subunit alpha (NED Millipore) was added to one of the tubes, while the control tube were added with glycerol. Both tubes were incubated at 30°C for 90 min. Flag-DLK was eluted from beads by boiling in the SDS sample buffer. Equal amounts of samples were analyzed by western blotting with anti-DLK and anti-pDLK$^{S302}$ antibodies.

## Statistical analysis

For experiments in flies, we knew from previous work that a sample size of 10 animals per genotype was large enough to detect significant differences among genotypes (*Xiong et al., 2010*; *2012*). Therefore, at least 10 animals ($\geq$ 50 axons) were examined and quantified in each genotype. Each experiment was repeated at least 3 times with independent biological replicates. For experiments in mice, we knew from previous work that measuring 100 neurons per genotype and an N=3–4 was sufficient to detect reproducible differences between the experimental groups (*Frey et al., 2015*; *Valakh et al., 2015*). Therefore, all the experiments were performed with at least 3 independent biological replicates.

One way ANOVA and multiple comparisons were conducted when more than two samples are compared. Tukey post-hoc test was used to correct for multiple comparisons. For binned DRG neurites length distribution, statistical significance was determined by two way ANOVA followed by Bonferroni post-hoc test. p values smaller than 0.05 were considered statistically significant. All p values are indicated as *p<0.05, **p<0.01, and ***p<0.001 and ****p<0.0001. Data are presented as mean $\pm$ SEM.

## Acknowledgements

We thank Dr. Joseph Lewcock, Dr. Daniel Kalderon and Dr. Stefan Strack for generous sharing of reagents. We thank Qiong Gao, Pushpanjali Soppina, Katharine Baldwin, and Rafi Kohen for technical support, and Haoxing Xu for sharing facilities. We thank the TRiP at Harvard Medical School (NIH/NIGMS R01-GM084947) for providing transgenic RNAi fly stocks. This work was supported by the National Institute of Health (grants NS069844 to CAC, and NS065053 to AD), Craig H Neilsen Foundation (AD), the Dr. Miriam and Sheldon G Adelson Medical Foundation on Neural Repair and Rehabilitation (RJG), Wings for Life Fellowship (CY), and F32 NS093962-01 (EF).

## Additional information

### Funding

| Funder | Grant reference number | Author |
|---|---|---|
| National Institute of Neurological Disorders and Stroke | F32 NS093962 | Erin Frey |
| Wings for Life | Postdoctoral Fellowship | Choya Yoon |
| Dr. Miriam and Sheldon G. Adelson Medical Research Foundation | | Roman J Giger |
| Craig H. Neilsen Foundation | | Aaron DiAntonio |
| National Institute of Neurological Disorders and Stroke | R01 NS065053 | Aaron DiAntonio |
| National Institute of Neurological Disorders and Stroke | R01 NS069844 | Catherine Collins |

The funders had no role in study design, data collection and interpretation, or the decision to submit the work for publication.

### Author contributions

YH, EF, CC, Conception and design, Acquisition of data, Analysis and interpretation of data, Drafting or revising the article; CY, Conception and design, Acquisition of data, Drafting or revising the article; HW, LBH, Conception and design, Contributed unpublished essential data or reagents; DN, Acquisition of data, Contributed unpublished essential data or reagents; RJG, AD, Conception and design, Drafting or revising the article

### Author ORCIDs

Roman J Giger, http://orcid.org/0000-0002-2926-3336
Catherine Collins, http://orcid.org/0000-0002-1608-6692

## Ethics

Animal experimentation: All mice and rats used in this study were housed and cared for in strict accordance with the recommendations in the Guide for the Care and Use of Laboratory Animals of the National Institutes of Health, and research conducted under the approval of the University of Michigan Medical School (UCUCA protocol #PRO00005896) and Washington University School of Medicine (UCUCA protocol #20150043) Committees on Use and Care of Animals.

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
