## [Decision Letter]

Thank you for submitting your work entitled "An evolutionarily conserved mechanism for cAMP elicited axonal regeneration involves direct activation of DLK" for consideration by *eLife*. Your article has been reviewed by three peer reviewers, and the evaluation has been overseen by Hugo Bellen as the Reviewing Editor and K VijayRaghavan as the Senior Editor.

The reviewers have discussed the reviews with one another and the Reviewing Editor has drafted this decision to help you prepare a revised submission.

Summary:

Hao et al. describe an evolutionarily conserved mechanism mediating cAMP axonal regeneration via the phosphorylation of dual leucine zipper-bearing kinase (DLK). DLK (known as Wnd in *Drosophila*) has previously been implicated in invertebrate and mammalian systems to be essential for neuronal axon regeneration. DLK is enriched in axons after axonal injury, and DLK is required for injury-induced axonal regeneration in neurons of mammals, flies and worms. However, the specific mechanism by which DLK is activated is currently unknown, but multiple signaling pathways are likely to activate axonal regeneration. The authors focus on the cAMP-PKA signaling pathway, which stimulates axonal elaboration in multiple models systems of axon growth/regeneration. Based on an evolutionarily conserved PKA phosphorylation site residing in the kinase domain of DLK, the authors hypothesized that DLK phosphorylation lies downstream of the cAMP-PKA pathway to modulate DLK-dependent axonal regeneration. Consistent with this hypothesis, the authors demonstrate that DLK is subject to direct PKA phosphorylation at the proposed PKA phosphorylation site (S302). Additionally, the authors demonstrate in HEK293 mammalian cultured cells that DLK phosphorylation is enhanced (phospho-specific DLK antibody) when the PKA signaling pathway is activated either pharmacologically or by expression of constitutively activated PKA. The authors correlate DLK phosphorylation with enhanced DLK abundance in HEK293 cells and implicate a previously described JNK feedback pathway that affects DLK ubiquitination and degradation. These observations are consistent with their well-executed in vivo experiments demonstrating #1) a genetic interaction in *Drosophila* between PKA and Wnd in mediating axonal regeneration after injury, #2) enhanced levels of Wnd protein and activity after axonal injury or genetically activating the PKA pathway, and #3) failure to rescue axonal regeneration by expressing the phosphorylation incapable Wnd S302A in the *wnd* mutant compared to WT controls.

PKA-phosphorylation of DLK/Wnd is a novel finding and of interest to the field of regeneration, unifying several previously described pathways into a coherent DLK-dependent pathway for injury-induced axon regeneration. Overall, the use of multiple model systems to describe DLK activation by PKA as a mechanism for axonal regeneration activation is one of the many strengths of this manuscript. Some comments for the authors to consider are noted below.

Essential revisions:

I recommend that the authors perform a few experiments to assess whether elevating cAMP levels in neurons mimics the expression of PKA^CA^ with regard to Wnd-dependent axonal sprouting and puckered induction. For instance, the authors could utilize loss-of-function alleles of the gene encoding the cAMP phosphodiesterase, *Dunce*. It will also be worthwhile to evaluate whether axon sprouting and puckered induction in dunce mutants are sensitive to Wnd knockdown. Conversely, the authors could examine if injury dependent axon sprouting and puckered induction are diminished if adenylyl cyclase (for e.g. Rutabaga or AC78C) levels are reduced.

Given that PKA activation leads to a strong upregulation of Wnd levels, I wonder whether Hiw and PKA function in the same pathway. I recommend that authors examine whether the increase in Wnd levels by PKA^CA^ is attenuated in the *hiw* mutants (by Westerns OR by imaging Wnd-GFP in axons). The results of this experiment will be interesting and relevant to this study regardless of the outcome! At the very least, the authors should discuss the stabilization of Wnd by PKA in the context of their previous findings regarding Hiw.

Is endogenous DLK/Wnd phosphorylated by PKA? The authors generated a beautiful DLK S302 phospho-specific antibody and successfully used it in transfected HEK293 cells expressing DLK. Can this antibody be used to demonstrate enhanced S302 phosphorylation of endogenous DLK/Wnd in *Drosophila* after expression of PKA^CA^ and/or axonal injury using westerns and/or immunostaining? Not sure if the antibody works across species, so this might not be an option. If not, has the antibody been tried in forskolin treated cultured cortical or DRG neurons? Would be a very nice positive addition to support the model, though not required given how difficult phospho-antibodies can be.

One point that could use some clarification for the readers is how the authors think that S302 phosphorylation actually affects DLK function. The authors nicely correlate DLK 302 phosphorylation with DLK abundance in transfected HEK293 cells. Huntwork-Rodriguez, in a prior study, reported that S302 of DLK is required for DLK protein stability (Huntwork-Rodriguez et al., 2013). That is, DLK downstream activation of the JNK pathway was proposed to result in a feedback loop in which JNK→DLK phosphorylation results in increased DLK abundance by reducing DLK degradation. There, the DLK^S302A^ mutant exhibited decreased overall DLK abundance by enhancing DLK degradation. Consistent with these previous findings, the current authors observed PKA-induced enhancement of DLK levels are blocked by JNK inhibitors without affecting S302 phosphorylation (Figure 3). Interestingly, the DLK^K185A^ kinase dead version of DLK remains enhanced AND phosphorylated at S302 with PKA activation even in the presence of JNK inhibitors (Figure 3), suggesting that elevation of DLK levels via S302 phosphorylation occurs *independent* of DLK kinase activity. How do the authors rationalize this in regards to their model (Figure 3) and previous studies? For example, Huntwork-Rodriguez et al. found that the S302A mutation abolished the ability of DLK to activate downstream effectors (JNK). This observation led directly to a previous model that DLK kinase activates JNK to feedback and phosphorylate DLK and regulate its abundance. If both the DLK^K185A^ and DLK^S302A^ mutants lack kinase activity and fail to activate downstream effector pathways (e.g. JNK), why do they behave differently with JNK inhibitors? The authors' statement that "The induction of DLK activation loop phosphorylation independently of its own signaling abilities places PKA firmly upstream of DLK, as an upstream regulator/activator” (Results) seems true, but what does this data mean for the proposed model (Figure 3)? Finally, does DLK^S302A^ mutant fail to rescue in the *wnd* mutant rescue experiments (Figure 3—figure supplement 1), as they fail to increase in general abundance in response to injury and/or are the basal levels of DLK^S302A^ lower compared to WT DLK?

The authors placed the WT and WndS302A *Drosophila* rescue experiments in the supplemental figures, but I'm not sure why these are not in the main text. As mentioned in #2, failure of the DLK^S302A^ rescues to enrich DLK in axons after PKACA expression or axon injury would strengthen the authors model that PKA-dependent phosphorylation of DLK modulates its local stability to promote regeneration programs. Finally, are the DRG experiments amenable to rescue experiments with WT and DLK^S302A^ rescue experiments?

Axons vs. cell body. PKA activity has much more effect on *wnd* levels in axons than cell bodies. Does this mean PKA is activated by injury in axons but not cell bodies? This should be possible to test with a PKA activity reporter, and possibly also the phospho-specific *wnd* antibody.

---

## [Author Response]

Essential revisions:

*I recommend that the authors perform a few experiments to assess whether elevating cAMP levels in neurons mimics the expression of PKA^CA^ with regard to Wnd-dependent axonal sprouting and puckered induction. For instance, the authors could utilize loss-of-function alleles of the gene encoding the cAMP phosphodiesterase, Dunce. It will also be worthwhile to evaluate whether axon sprouting and puckered induction in dunce mutants are sensitive to Wnd knockdown. Conversely, the authors could examine if injury dependent axon sprouting and puckered induction are diminished if adenylyl cyclase (for e.g. Rutabaga or AC78C) levels are reduced.*

In our revision we have included new data with knockdown of phosphodiesterase *Dunce* in *Drosophila* motoneurons, which leads to enhanced axonal sprouting response (Figure 1). These results are consistent with previously published data in *C. elegans* (Ghosh-Roy et al., 2010). Note that this previous study already showed that the enhanced regeneration in phosphodiesterase mutants required dlk-1 function.

We have also performed the puckered reporter analysis. We found that knockdown of Dunce in *Drosophila* motoneurons cause increased puckered expression in uninjured animals (Figure 2), which is predicted from our model that cAMP activates DLK/Wnd signaling.

Given that PKA activation leads to a strong upregulation of Wnd levels, I wonder whether Hiw and PKA function in the same pathway. I recommend that authors examine whether the increase in Wnd levels by PKA^CA^ is attenuated in the hiw mutants (by Westerns OR by imaging Wnd-GFP in axons). The results of this experiment will be interesting and relevant to this study regardless of the outcome! At the very least, the authors should discuss the stabilization of Wnd by PKA in the context of their previous findings regarding Hiw.

We agree that the relationship between Hiw and PKA is very interesting to understand, particularly since some biochemical data in other species suggest that Hiw/Pam may regulate cAMP levels by inhibiting adenylate cyclase activity (Ehnert et al., 2004) (Pierre et al., 2004). Our data thus far suggest that the relationship is complex, so including data/experiments on this issue would be both distracting to the current story, and would require many more experiments to address the relationship properly. So instead we are following the reviewer’s suggestion to discuss the regulation of Wnd by PKA in the context of what is previously known for Hiw, (in the Discussion section).

Is endogenous DLK/Wnd phosphorylated by PKA? The authors generated a beautiful DLK S302 phosphospecific antibody and successfully used it in transfected HEK 293 cells expressing DLK. Can this antibody be used to demonstrate enhanced S302 phosphorylation of endogenous DLK/Wnd in Drosophila after expression of PKACA and/or axonal injury using westerns and/or immunostaining? Not sure if the antibody works across species, so this might not be an option. If not, has the antibody been tried in forskolin treated cultured cortical or DRG neurons? Would be a very nice positive addition to support the model, though not required given how difficult phospho-antibodies can be.

We have tested the pS302-DLK antibody in flies and unfortunately it does not work across species. Over the past two months, we have tried hard to get a decent signal for endogenous DLK in cultured neurons, including scaled up cultures and immunoprecipitations to enrich for the protein. This has been very challenging technically – we think that the levels of endogenous DLK are very low, and that the antibodies may also not be very sensitive. We prefer to leave this experiment out of the paper. As the phosphorylation of S302 is important for DLK to function as a kinase, our functional data combined with our biochemical data in HEK293 cells strongly suggests that S302 is indeed phosphorylated on endogenous DLK.

One point that could use some clarification for the readers is how the authors think that S302 phosphorylation actually affects DLK function. The authors nicely correlate DLK 302 phosphorylation with DLK abundance in transfected HEK293 cells. Huntwork-Rodriguez, in a prior study, reported that S302 of DLK is required for DLK protein stability (Huntwork-Rodriguez et al., 2013). That is, DLK downstream activation of the JNK pathway was proposed to result in a feedback loop in which JNK→DLK phosphorylation results in increased DLK abundance by reducing DLK degradation. There, the DLK^S302A^ mutant exhibited decreased overall DLK abundance by enhancing DLK degradation. Consistent with these previous findings, the current authors observed PKA-induced enhancement of DLK levels are blocked by JNK inhibitors without affecting S302 phosphorylation (Figure 3). Interestingly, the DLK^K185A^ kinase dead version of DLK remains enhanced AND phosphorylated at S302 with PKA activation even in the presence of JNK inhibitors (Figure 3), suggesting that elevation of DLK levels via S302 phosphorylation occurs independent of DLK kinase activity. How do the authors rationalize this in regards to their model (Figure 3) and previous studies? For example, Huntwork-Rodriguez et al. found that the S302A mutation abolished the ability of DLK to activate downstream effectors (JNK). This observation led directly to a previous model that DLK kinase activates JNK to feedback and phosphorylate DLK and regulate its abundance. If both the DLK^K185A^ and DLK^S302A^ mutants lack kinase activity and fail to activate downstream effector pathways (e.g. JNK), why do they behave differently with JNK inhibitors? The authors' statement that "The induction of DLK activation loop phosphorylation independently of its own signaling abilities places PKA firmly upstream of DLK, as an upstream regulator/activator” (Results) seems true, but what does this data mean for the proposed model (Figure 3)?

We thank the reviewers for these points. We have carried out and included some additional experiments (which lead to an expansion of the previous Figure 3 into two figures: Figure 3 and Figure 4). These data provide further clarification and lead to a revised model (Figure 4). Both the S302A and K185A mutants appear to be ‘kinase dead’ since they are unable to stimulate downstream signaling, including phosphorylation of JNK and c-Jun (Huntwork-Rodriguez et al., 2013). We found that for both mutant proteins, PKA activation stimulates an increase in DLK levels (Figure 4), even though they are incapable of downstream signaling. This suggests that PKA enhances DLK stability via an independent mechanism that is separate from the previously described JNK-dependent feedback mechanism. In further support of this, JNK inhibition acts additively with PKA activation (Figure 4). This revised model (in Figure 4) unifies our observations with previous studies (including Huntwork-Rodriguez et al) and has some additional interesting implications for the regulation of DLK, which we have discussed further in this revised manuscript.

Finally, does DLK^S302A^ mutant fail to rescue in the wnd mutant rescue experiments (Figure 3—figure supplement 1), as they fail to increase in general abundance in response to injury and/or are the basal levels of DLK^S302A^ lower compared to WT DLK

Since independent transgenes have different expression levels, we cannot fairly assess the effect of the mutation upon basal expression levels. Using anti-Wnd antibodies to detect the over-expressed Wnd protein, we have examined basal and injury induced Wnd levels by immunocytochemistry. The transgenic protein is still quite hard to detect, and we do not observe an induction in Wnd^S301A,305A^ mutant protein after axonal injury. But we have caveats to reach any firm interpretations with this data, since we know that the expression levels of the mutant transgene is extremely high compared to endogenous Wnd.

The authors placed the WT and WndS302A Drosophila rescue experiments in the supplemental figures, but I'm not sure why these are not in the main text. As mentioned in #2, failure of the DLK^S302^A rescues to enrich DLK in axons after PKACA expression or axon injury would strengthen the authors model that PKA-dependent phosphorylation of DLK modulates its local stability to promote regeneration programs.

We appreciate this suggestion and have moved the rescue data into the main figures (Figure 3). As discussed above and in our revised manuscript, the fact that this mutant protein is non-functional is important to emphasize since the stabilizing effect of PKA still occurs for the DLK^S302A^ mutant.

*Finally, are the DRG experiments amenable to rescue experiments with WT and DLK^S302A^ rescue experiments?*

We would like to do such rescue experiments. Unfortunately, they are not very feasible in adult DRG neurons. Both transfection and infection is very inefficient in adult DRG neurons for constructs the size of DLK. In addition, overexpression of DLK in neurons leads to DLK auto-activation and induces apoptosis.

Axons vs. cell body. PKA activity has much more effect on wnd levels in axons than cell bodies. Does this mean PKA is activated by injury in axons but not cell bodies? This should be possible to test with a PKA activity reporter, and possibly also the phospho-specific wnd antibody.

We agree that it is very interesting to know whether PKA regulates DLK locally in axons, however this is currently a very technically challenging question. The phospho-DLK antibodies do not work in flies, and our recent efforts lead us to conclude that they are also intractable for IHC in mammalian neurons (due to a high degree of non-specific staining). While a FRET reporter has been used for PKA in flies (Gervasi et al., 2010), it has never been used before in our larval motoneuron preparation. We have begun to work with a cAMP FRET reporter that is more widely used in *Drosophila* (Ponsioen et al., 2004) and still have a great deal of troubleshooting to carry out in order to implement the reporter in a new preparation and verify that it is working properly. The PKA reporter would require even further validation and controls. We think this is an interesting future project, but is not practical for this current story. An important additional caution is that even if any of these suggested assays work, a negative result in our injury assays could still have the caveat that the method of detection is not sensitive enough to detect functional changes that indeed occur.